# The Contribution of the Nrf2/ARE System to Mechanotransduction in Musculoskeletal and Periodontal Tissues

**DOI:** 10.3390/ijms24097722

**Published:** 2023-04-23

**Authors:** Athanassios Fragoulis, Mersedeh Tohidnezhad, Yusuke Kubo, Christoph Jan Wruck, Rogerio Bastos Craveiro, Anna Bock, Michael Wolf, Thomas Pufe, Holger Jahr, Frank Suhr

**Affiliations:** 1Department of Anatomy and Cell Anatomy, Uniklinik RWTH Aachen, RWTH Aachen University, 52074 Aachen, Germany; mtohidnezhad@ukaachen.de (M.T.); ykubo@ukaachen.de (Y.K.); cwruck@ukaachen.de (C.J.W.); tpufe@ukaachen.de (T.P.); hjahr@ukaachen.de (H.J.); 2Department of Orthodontics, Dental Clinic, Uniklinik RWTH Aachen, RWTH Aachen University, 52074 Aachen, Germany; rcraveiro@ukaachen.de (R.B.C.); michwolf@ukaachen.de (M.W.); 3Department of Oral and Maxillofacial Surgery, Uniklinik RWTH Aachen, RWTH Aachen University, 52074 Aachen, Germany; abock@ukaachen.de; 4Institute of Structural Mechanics and Lightweight Design, RWTH Aachen University, 52062 Aachen, Germany; 5Division of Molecular Exercise Physiology, Faculty of Life Sciences: Food, Nutrition and Health, University of Bayreuth, 95326 Kulmbach, Germany

**Keywords:** Keap1, Nrf2, mechanobiology, reactive oxygen species, mechanosensitive tissue, oxidants, redox signaling

## Abstract

Mechanosensing plays an essential role in maintaining tissue functions. Across the human body, several tissues (i.e., striated muscles, bones, tendons, ligaments, as well as cartilage) require mechanical loading to exert their physiological functions. Contrary, mechanical unloading triggers pathological remodeling of these tissues and, consequently, human body dysfunctions. At the cellular level, both mechanical loading and unloading regulate a wide spectrum of cellular pathways. Among those, pathways regulated by oxidants such as reactive oxygen species (ROS) represent an essential node critically controlling tissue organization and function. Hence, a sensitive balance between the generation and elimination of oxidants keeps them within a physiological range. Here, the Nuclear Factor-E2-related factor 2/Antioxidant response element (Nrf2/ARE) system plays an essential role as it constitutes the major cellular regulation against exogenous and endogenous oxidative stresses. Dysregulations of this system advance, i.a., liver, neurodegenerative, and cancer diseases. Herein, we extend our comprehension of the Nrf2 system to the aforementioned mechanically sensitive tissues to explore its role in their physiology and pathology. We demonstrate the relevance of it for the tissues’ functionality and highlight the imperative to further explore the Nrf2 system to understand the physiology and pathology of mechanically sensitive tissues in the context of redox biology.

## 1. Introduction

Defined tissues own the capability to sense and respond to mechanical load, a process that is known as mechanotransduction [1]. Mechanisms of mechanotransduction ensure the physiological properties of a variety of musculoskeletal tissues, including skeletal muscles, cartilage, and ligaments, as well as bones and tendons [2,3,4,5,6,7]. Contrary, the loss of physiological loading, a condition called unloading, provokes pathological and malfunctional consequences, finally resulting in the deterioration of musculoskeletal tissues [2,5].

Mechanotransduction processes link extracellular cues via ion channels, G-protein coupled receptors, growth factor receptors, as well as integrins and integrin-related signaling pathways to intracellular mechanisms that crucially control and maintain cellular homeostasis and, hence, tissue and organ integrity [2,3]. Among the myriad of cellular processes concertedly controlling cellular functions, the nuclear factor Erythroid 2-related factor 2 (Nrf2)—antioxidant/electrophile response element (ARE/EpRE) system exerts pivotal roles in the physiological maintenance of cells and consequently of tissues.

The transcription factor Nrf2 has widely been described as one of the main transcription factors involved in the maintenance and adaptation of intracellular redox homeostasis [8,9]. It is important to point out, however, that in this context, biological redox equilibria should not be understood as thermodynamic equilibria but rather as a non-equilibrium within defined boundaries, what is commonly referred to as a “steady state” [10]. The emergence and/or production of oxidants within cells leads to a shift of the intracellular redox status toward oxidative conditions. When this shift, however, remains within these physiological boundaries, this can be referred to as oxidative eustress. Since these conditions do not lead to damage of biomolecules but rather specific redox-sensitive protein targets are regulated, oxidative eustress plays a crucial role in redox control and physiological redox signaling [11,12,13,14]. In contrast, oxidative conditions that exceed these natural limits should be instead referred to as oxidative distress, which ultimately can cause serious cellular damage [12]. Such a biphasic dose-response phenomenon nicely corresponds to a concept better known in the field of toxicology called hormesis [15], in which a stimulatory response occurs at a low dose and an inhibitory response at a high dose. As nicely reviewed by Sies and Ursini just recently, Nrf2 exerts its primary function under physiological ranges of stress exposure meaning conditions of oxidative eustress to restore the “golden mean” and is thus essential for functional redox signaling [9]. This highlights that pro-oxidative shifts are absolutely essential for the physiology and maintenance of a cell or tissue. Therefore, the occurrence and/or production of endogenous oxidants, which also but not exclusively include reactive oxygen species (ROS), is not harmful per se, but fulfills important physiological functions. To do so, Nrf2 regulates the transcription of a variety of genes encoding for cytoprotective and antioxidant enzymes and thereby orchestrates the redox homeostasis.

The transcriptional activity of Nrf2 is mediated by a highly conserved DNA sequence located in the promoter region of its target genes. This sequence is referred to as the antioxidant or electrophile response element (ARE/EpRE) [16,17,18,19]. As also known for other stress-responsive factors, Nrf2 is constitutively and ubiquitously expressed, while its regulation mainly takes place at the protein level (Figure 1A). Currently, there are two mechanisms known by which Nrf2 activity is negatively regulated. In the original redox-sensitive canonical pathway, the adapter protein Kelch, similar to the ECH associate protein 1 (Keap1), plays the central role, binding Nrf2 and thereby subjecting it to Cullin3/Rbx-mediated polyubiquitination [20,21]. In addition, Nrf2 is regulated by the non-canonical pathway, in which GSK3-β-mediated phosphorylation in the region of a degron causes the adapter molecule β-TrCP to bind to Nrf2. This binding results in assembly with a Cullin1/Rbx complex and, thus, also in polyubiquitination [22]. Both pathways consequently lead to proteosomal degradation of Nrf2. Whereas Keap1-dependent Nrf2 activation is mediated primarily by oxidants and electrophiles, GSK3-β inhibition mainly results from the activation of a variety of growth factor receptors. This provides an additional level of redox-independent control of Nrf2 activity, by which growth factor signaling such as the PI3K/Akt axis can activate this system. However, since downstream of some of these receptors, the production of oxidants is also involved, this may as well promote Keap1-dependent Nrf2 activation in parallel (Figure 1B). Moreover, it has been shown that the activation of downstream MAP kinases as a result of receptor binding can activate Nrf2 as well [23,24]. This already clearly indicates that there obviously are functions of Nrf2 beyond those of pure oxidative stress defense. Over the last two decades, research revealed convincing evidence for such Nrf2 functions that demonstrate its involvement in many different cellular processes such as metabolism [25], cell cycle regulation [26,27], regeneration [28,29,30], and beyond.

A possible link between the mechanotransduction machinery and the Nrf2/ARE system has been observed in musculoskeletal tissues. For instance, researchers found that chronic mechanical stimulation of skeletal muscles through exercise interventions boosts the antioxidant defense capacities by increasing and subsequently activating the Nrf2/ARE system [31]. As another example, bones demonstrate high resistance against mechanical forces, and their physiological capabilities rely on those. Comparable to skeletal muscles and tendons, bones benefit from an elevated and active Nrf2/ARE system to maintain their physiological capacities, including growth, remodeling, bone mineral density, as well as regeneration upon injury. In this regard, inactivation of the Nrf2/ARE system promotes differentiation as well as activation of osteoclasts, an event that triggers bone resorption [32,33]. Collectively, these examples highlight the relevance of the proper integrity of the Nrf2/ARE system in mechanically sensitive tissues and point to the assumption that disturbances of this antioxidant defense system cause tissues’ dysfunctions, independent of the dysregulation of the reactive oxidant species homeostasis.

As the general Nrf2 activation mechanisms and the currently known signaling cascades involved in mechanotransduction share common actors (Figure 2), the aim of this review is to search the current literature and find evidence for this potential link between Nrf2 and mechanotransduction in the musculoskeletal system.

We focus on and highlight the relevance of this master antioxidant defense regulator in the physiology and pathology of mechanosensitive musculoskeletal tissues and organs such as striated muscles, bones, tendons, and cartilage to express the significance of making further efforts to decode the cellular mechanisms controlled by the Nrf2 system ensuring cellular and tissue health.

## 2. Mechanobiology and the Nrf2/ARE System in Striated Muscles

Striated muscles comprise skeletal and cardiac muscle characterized by their prominent cross striations defined by the Z-discs of the sarcomeres, which reflect the contractile units of skeletal and cardiac muscle tissues. Mechanical loading is essential for the maintenance and functionality of both skeletal and cardiac muscles. For instance, chronically reduced mechanical load causes loss of skeletal muscle tissue and its structural as well as metabolic dysfunction [34,35,36].

Mechanical and metabolic properties of skeletal muscles rely on the skeletal muscle fiber types defined as slow-twitch/oxidative (known as type 1) and fast-twitch/glycolytic (known as type 2A, type 2X, and type 2B) fiber types represented by their predominant myosin-heavy chain isoform. Human skeletal muscles demonstrate three myosin-heavy chain isoforms, type 1, type 2A, as well as type 2X [37]. In contrast, mouse skeletal muscles comprise an additional myosin-heavy chain isoform, namely type 2B, which indicates that mice have four different skeletal muscle fiber types in total [37]. Importantly, cardiac muscles comprise a single myosin-heavy chain, the type 1 or β-myosin isoform [37].

Skeletal and cardiac muscles comprise defined networks of proteins that transform the physical, mechanical cue into biological signals. These protein assemblies, known as costameres or focal adhesions, link the extracellular matrix (ECM) microenvironment to striated muscle-intrinsic structures and locate beneath the sarcolemma in the closest vicinity of integrin receptors to properly “sense” the mechanical load and to trigger a defined downstream signaling pathway (reviewed in [1,3,38,39]). In human and murine skeletal muscles, components of the costamere network demonstrate skeletal muscle fiber type-specific localization [1,4,6], which sheds new light on the role and mechanistic functions of costamere components in skeletal muscles. In cardiac muscle, a new costamere component, known as β-parvin, controls cardiac muscle development, contractile forces, and sarcomerigenesis through a Rac1-dependent mechanism [40], demonstrating that the mechanosensitive network and signaling cues are far from being understood in striated muscles.

In this context, the Nrf2 system is fascinating since it seems to possess crucial functions in the maintenance of striated muscle functionality. In healthy skeletal muscles subjected to increased mechanical load and accompanied by enhanced metabolic flux as provoked by exercise, Nrf2 gets activated to control for intracellular redox signaling as oxidants, such as reactive oxygen species (ROS), rise during and upon exercising conditions. Hence, under physiological conditions, Nrf2 indirectly controls mechanical loading conditions by priming skeletal muscle fibers to adjust their antioxidant defense capacities. Contrary, sarcopenia, which defines an aging-related progressive and generalized skeletal muscle disorder characterized by an accelerated loss of muscle mass and function, results in chronically elevated levels of oxidants paralleled by a decreased antioxidant defense capacity [41], which leads to oxidative distress and associated tissue destruction. Hence, it was hypothesized that aging results in a steady decline in Nrf2 bioavailability accompanied by increased oxidative distress and mitochondrial dysfunction, which is a consequence of sarcopenia and one of its driving mechanisms. Regarding this assumption, it was shown that Nrf2 availability indeed decreases with aging in murine skeletal muscle. Using a *Nrf2* knockout model, the same research group additionally observed that *Nrf2* deficiency causes physical dysfunction and increased frailty phenotypes paralleled by reduced mitochondrial density and biogenesis [42]. This study convincingly identified Nrf2 as an essential factor in maintaining skeletal muscle integrity and functionality during aging. In the context of aging, Nrf2 possesses auxiliary-crucial functions in skeletal muscle regeneration. *Nrf2* deficiency causes reduced regeneration capacities in aged skeletal muscles through impaired Pax7/MyoD expressions representing lower activities of skeletal muscle stem cells, the so-called satellite cells [43]. Importantly, skeletal muscle aging involves a balance between oxidants and antioxidants, and exercise has been proven to regulate the balance of oxidants to ameliorate the process by which skeletal muscle fibers might protect themselves against these stressors by improving their intrinsic antioxidant capacities. Interestingly, Nrf2 possesses skeletal muscle regeneration-regulating capacities apart from its canonical redox capabilities. In a skeletal muscle ischemia–reperfusion injury model, our group demonstrated that *Nrf2* deficiency results in impaired skeletal muscle regeneration due to reduced *MyoD* expression [44], a crucial satellite cell proliferation manager. Hence, Nrf2 activity can be interpreted as a prerequisite to guide skeletal muscle regeneration, independent of its antioxidant defense-governing functions.

Another example of detrimental skeletal muscle mass loss is known as cancer cachexia. About 80% of people with various cancer types suffer from serious skeletal muscle mass loss. Importantly, about 30% of these cancer patients die from this severe skeletal muscle mass loss. Compared to normal, healthy cells, cancer cells increase their oxidant production. In cancer cells and cancer growth, these possess dual roles. On the one hand, they foster protumorigenic signaling, hence, cancer cell proliferation; on the other hand, they provide antitumorigenic signaling that results in oxidative stress-induced cancer cell death [45]. Consequently, and to withstand high oxidant levels, cancer cells increase their antioxidant capacities which finally guarantees their survival–a process called adaptive homeostasis [46]. Interestingly, the Nrf2 system, in its function as the master regulator of antioxidant defense, has not been investigated in the context of skeletal muscle mass wasting during cachexic conditions. One recent study addressed the role of Nrf2 in immortalized C2C12 myotubes by activating Nrf2 signaling through the administration of sulforaphane. Sulforaphane serves as a potent Nrf2 activator [47], wherefore the authors hypothesized that C2C12 myotube treatment with sulforaphane could prevent C2C12 myotube shrinking. The authors indeed found that a sulforaphane treatment induced C2C12 myotube hypertrophy and myotube fusion through ERK signaling. C2C12 myotubes cocultured with Colon-26 cancer cells demonstrated reduced cell mass in the absence of sulforaphane treatment, which was finally attenuated by sulforaphane administration to this cell culture model [48]. Although too early to conclude from this in vitro study, it can be assumed that Nrf2 might exert relevant control core with an essential role in preventing skeletal muscle mass loss during cancer cachexia episodes. To get a clearer picture on this relevant question, sophisticated in vivo studies have to prove this assumption.

Among aging- and cancer cachexia-related skeletal muscle mass loss, prolonged inactivity provokes similar effects. An excessive condition of prolonged inactivity is space flight associated with predominantly type 1 fiber-related skeletal muscle mass loss and accompanied by a type 1 to type 2 skeletal muscle fiber type transition. The role of Nrf2 in space flight-mediated skeletal muscle mass loss is largely unknown. However, recently an elegant study shed light on this relevant question. In this study [49], the authors sent wild-type and *Nrf2* knockout mice to the International Space Station for 31 days. They observed similar mass loss of soleus muscles in both wild-type and *Nrf2* knockout mice. However, the 31-day-lasting space accommodation caused an accelerated type 1 to type 2A skeletal muscle fiber type transition. Hence, Nrf2 seems to possess a relevant function in the regulation of skeletal muscle fiber transitions, a result relevant to understand mechanisms that induce skeletal muscle fiber-type transitions under conditions of skeletal muscle mass loss.

In analogy to skeletal muscles, antioxidant defense systems play crucial roles in the maintenance of cardiomyocytes. For instance, cardiomyopathies go along with significantly increased oxidant levels that exacerbate these clinically relevant conditions. Consequently, increased levels of antioxidant enzymes could play crucial roles in ameliorating cardiomyopathies and cardiac diseases in general. Comparable to skeletal muscles and basically all cell types, cardiomyocytes are sensitive to alterations in the redox state, which means that shifts in their redox environments potentially stimulate cardiac disease phenotypes. Nrf2 has been hypothesized to prevent or at least ameliorate cardiac diseases through its capability to increase the cardiomyocyte’s antioxidant defense. Interestingly, researchers found that Nrf2 deficiency could reflect a condition that prevents cardiomyopathies. Using a cardiac disease mimicking mouse model developing a mutant protein aggregation cardiomyopathy (MPAC) caused by mutations in the protein alpha B-Crystallin, the authors demonstrate that normal *Nrf2* expression in the MPAC model deteriorated the cardiac phenotype reflected by heart failure, whereas the MPAC model with *Nrf2* deficiency did not develop heart failure. Mechanistically, *Nrf2* deficiency in the MPAC model suppressed reductive stress, the opposite of oxidative stress, and hence reduced cardiac protein aggregation and consequently improved the survival of the mice [50].

In addition, different studies explored the potential of nutrigenomic Nrf2 activators, such as sulforaphane, curcumin, quercetin, or polyphenols, in the maintenance of cardiac health. It was found that grapes possess the potential to reduce hypertension, a heart failure-promoting disorder affecting about one billion people worldwide. Mechanistically, grapes activated antioxidant defense pathways that increased the activity of genes, i.e., *Nrf2*, boosting the production of glutathione [51]. Another mechanism through which nutrigenomic compounds contribute to cardiac and, hence, human health is the induction of Nrf2 protein nuclear shuttling. Under normal conditions, Nrf2 oscillates between the nucleus and the cellular cytoplasm every 129 min. However, nutrigenomic activation of Nrf2 caused a more frequent oscillation of 80 min between the nucleus and the cytoplasm [52]. To provide another example, Nrf2 activation through sulforaphane prevented metabolic dysfunctions in human microvascular endothelial cells under hyperglycemic conditions, a situation present in Diabetes Mellitus type 2 patients and that is characterized, i.a., by increased oxidant levels. Specifically, the authors demonstrated that Nrf2 activation through sulforaphane caused Nrf2 nuclear translocation accompanied by increased ARE-linked gene expression, such as transketolase and glutathione reductase. In addition, hyperglycemia increased ROS levels in human microvascular endothelial cells, which was rescued by the administration of sulforaphane. Furthermore, the authors showed that *Nrf2*-deficient human microvascular endothelial cells exhibit higher ROS levels [53], further pointing to a relevant role of Nrf2 in Diabetes Mellitus type 2-related cardiovascular diseases.

Overall, these data demonstrate that Nrf2 possesses pivotal roles in the health maintenance of the mechanosensitive striated muscle tissues. However, the defined link between mechanosensing pathways and the Nrf2-Keap1 system still has to be elucidated in detail, whereas this link provides a promising conjunction to further understand the potential of mechanosensing components on the regulation of antioxidant defense properties of striated muscles.

## 3. Mechanobiology and the Nrf2/ARE System in Periodontal Alveolar Remodeling

The periodontal tissue is a complex connective structure composed of diverse cell types that continually interact to maintain alveolar bone, gingiva, cementum, and periodontal ligament (PDL) [54]. These tissues, composed of diverse mineralized and non-mineralized structures, constitute the periodontium [55]. The PDL is closely attached to the alveolar bone and cementum, which in turn connects the tooth to the bone socket. PDL is a fibrous tissue composed of heterogeneous cell populations (fibroblasts, cementoblasts, osteoclasts, osteoblasts, osteocytes, and endothelial cells) and non-cellular components (mostly from collagen fibers and non-collagenous ECM proteins and polysaccharides) that are all intertwined with the nerve fibers and blood vessels [56]. The periodontal ligament serves as a cushion for the mechanical stimuli created during mastication and provides a protective barrier to the microorganisms in the oral cavity [57,58].

The periodontium with alveolar bone creates a unique structure. Nowhere else in the body does a soft tissue such as PDL have a comparable ability to regulate hard tissue (alveolar bone) remodeling due to mechanical stress (i.e., shear, tension, and compression) [59].

The alveolar bone lines the alveolus, which supports dentition, i.e., other than the basal bone, which is an osseous tissue of the mandibula and maxilla [60]. Furthermore, the basal bone is not involved in the alveolar process. The alveolar process rests on the basal bone of the mandible and maxilla and consists of the alveolar bone, cortical plate, and sponge bone. The alveolar process is the bone that possesses teeth and *alveoli dentales*. The alveolar process and basal bone are located beside each other, and there is no clear separation between them. The alveolar bone undergoes extensive remodeling according to tooth movement and external stimuli. However, the presence and maintenance of the alveolar bone are tooth-dependent. Following the loss of a tooth due to a trauma or tooth extraction, the alveolar bone tends to resorb.

The remodeling in the PDL is controlled by two different regulative processes, both of which have inflammatory reactions in the background but are induced either by bacteria or non-bacterial stimuli (so-called sterile inflammation). It is essential to differentiate the origins of these inflammatory reactions, although they can occur simultaneously in some cases [59,61,62].

Clinical evidence indicates that the continuous remodeling of the tooth root cementum and periodontal apparatus in response to mechanical stimulation maintains tissue strength, prevents tissue damage, and secures teeth anchorage. The tooth root cementum and the periodontal ligament are the key regulators of tissue homeostasis within the tooth surrounding tissues. The root cementum has the ability to prevent the tooth root from damage and supplies the anchorage of the periodontal ligament to the tooth root. The periodontal ligament control tissue resorption, reorganization in frequent tissue remodeling processes, and transmission of mechanical signals [63,64,65].

Through mastication, orthodontic tooth movement, and trauma, mechanical stimuli are generated [57,63], influencing mechanosensory cells of the PDL that is closely attached to the alveolar bone and cementum, which in turn is attached to the tooth causing local hypoxia and fluid flow and initiating a sterile inflammation cascade enabling bone remodeling [61,66]. These changes include activation of cellular signaling cascades, differential gene and protein expression [67,68], and release of factors that modulate local inflammation together with differentiation programs of diverse cells regulating alveolar bone remodeling and health [69,70]. By orthodontic tooth movement, bone resorption occurs on the compression side with resorption of the alveolar bone and degradation of the periodontal ligament, while new bone is formed on the tension side with stretching of the periodontal ligament-inducing bone apposition and alignment of the Sharpey fibers [71,72,73,74].

PDL cells are known to have a strong impact on local osteoclastogenesis during mechanical loading. This impact results in an enhanced expression of osteogenic markers, such as RunX2, Osterix, and OPG, and a corresponding downregulation of osteoclastogenic markers, such as RANKL [75]. In contrast, when mechanical loading is absent, PDL cells attract osteoclast precursors via ICAM1 presentation and support the expression of osteoclastogenesis-beneficial molecules, such as RANKL, MCSF, and TNF-α [65].

Up to now, there is only minor knowledge about Nrf2′s role in the PDL under mechanical loading. However, the group of Xi and colleagues addressed this issue in the last two years and were able to bring some light to this topic. With respect to their studies, Nrf2 seems to play an important role in periodontal remodeling, especially in the osteogenic differentiation of periodontal ligament stem cells (PDLSCs) under mechanical stress. In 2021, they demonstrated that the cyclic mechanical stretch of PDLSCs induced osteogenic differentiation while simultaneously inducing Nrf2 activity and the expression of typical Nrf2 target genes. They showed that an increased hydrogen peroxide (H_2_O_2_) and superoxide anion (O_2_^•−^) production within the cells could be a valid explanation for the observed Nrf2 activation. *Nrf2* silencing via siRNA delivery resulted in reduced osteogenic differentiation of these cells [76]. In a follow-up study, the same group investigated the underlying mechanisms that could be responsible for the mechanical stretch-mediated Nrf2 activation in more detail in vitro as well as in vivo. They have shown that in addition to their previous findings, the observed Nrf2 activity under cyclic mechanical stretch was also induced via the PI3K/Akt signaling pathway—a pathway that evidently activates Nrf2. With these two studies, Xi et al. provide a first link between mechanotransduction via oxidants as well as PI3K- signaling (Figure 2) the Nrf2 pathway as well as its importance for osteogenic differentiation of the PDL [77]. However, this relationship is still not clarified in detail yet, as in the same year, another study of this group was published, showing that the administration of N-acetylcysteine (NAC) during cyclic mechanical loading enhances osteogenic differentiation by decreasing Nrf2 activity. This inhibition of Nrf2 activity by NAC had beneficial effects on the microstructure of the alveolar bone and enhanced the expression levels of osteogenesis markers in PDL in orthodontic rats on the tension side [78].

Taken together, these data demonstrate that the PDL organizes mechanical cues in the development of and maintenance of alveolar bone structures. It has already been shown, at least to some extent, that the Nrf2 system plays a key role in this process (Figure 3). The currently available data shows that the Nrf2-system controls the process of periodontal alveolar remodeling, which defines it as a crucial control hub for ECM remodeling in the periodontal alveolar structures under mechanical load. However, due to the limited number of studies that have addressed this topic, this assumption definitely needs to be thoroughly confirmed in future research.

## 4. Mechanobiology and the Nrf2/ARE System in Bones

Bone is a dynamic tissue that undergoes continuous remodeling and plays an important role in the body through various physiological functions, including (but not limited to) calcium and phosphate ion storage, blood cell neogenesis in the bone marrow, or simply mechanical protection of internal organs [79]. In addition, the coordinated activity of bone formation and bone resorption provides a mechanism for bone remodeling [79]. Furthermore, various hormones, as well as the nervous system, regulate bone homeostasis. Finally, bone cells, specifically osteocytes, possess the capability to sense and respond to mechanical stimuli, a hallmark characteristic controlling bone health resorption [80,81,82].

For instance, a major influence on the bony remodeling of the jaw is caused by mastication. These tactile stimuli are transmitted to the central nervous system by mechanoreceptors of periodontal ligaments (PDL) [83].

Osteocytes constitute the majority of bone-representing cells, reside within the mineralized bone matrix, and form a mesh-like intercellular network with long cell protrusions. Structurally, osteocytes are embedded in the lacuno–canalicular system (LCS), and a pericellular matrix (PCM) distinct from the mineralized ECM surrounds the lacunar osteocytes. The mechanical microenvironment ECM is predominately constituted of fibrillary collagens and transmits physical signals to the osteocyte cytoskeleton [84].

As a result of these mechanically controlled signals, a variety of intracellular signaling cascades are activated. Similar to cells in other tissues, Nrf2 is also involved in the protection and maintenance of osteocyte function. Nrf2 activity has been reported to increase the expression of osteocyte-specific markers, such as dentin matrix protein-1 (Dmp1), matrix-extracellular phosphoglycoprotein (Mepe), and sclerostin (Sost) in vivo and in vitro. Osteocyte-specific *Nrf2* deletion resulted in osteopenia. In line with that finding, it was shown that the administration of the Nrf2 inducer dimethyl fumarate (DMF) prevented detrimental effects on trabecular bone in ovariectomized mice and restored gene expression of decreased osteocyte-specific markers [85]. Controversial to this, Yoshida and colleagues showed that a genetic Nrf2 hyperactivation using a global Keap1 knockout led to hypoplasia due to impaired osteoblast differentiation [86]. Moreover, regarding bone acquisition, it should be critically reported that in addition to the Nrf2 status, age, as well as sex, strongly impact bone parameters as demonstrated using Nrf2 knockout models. Especially in female mice, there are reports of decreased bone mineral density/bone mass and osteoporotic changes associated with aging [87,88,89]. Delayed fracture repair in Nrf2-deficient mice has also been reported [90,91]. Remarkably, mechanical stimulation through running exercise effectively recovered Nrf2 activity and downstream expression of Nrf2 target genes in ovariectomized mice. Importantly, these data show that Nrf2 was essential to maintain femur bone mineral density (BMD) [92]. Therefore, the elucidation of the Nrf2 pathway in mechanically sensitive osteocytes might be a promising objective to pave the way toward the maintenance of bone mineral density.

In conclusion, the Nrf2-Keap1 system is involved in the activation and cell defense of osteocytes, mechanosensors in bone. It has the potential to lead not only to the protection of bone metabolism under oxidative stress (therapeutic effect for osteoporosis and intractable fractures) but also to the recovery of the physiological function of bone in the whole organism or to contribute to daily physical activities as a mechanosensor.

## 5. Mechanobiology and the Nrf2/ARE System in Tendons

Tendon and ligament injuries are responsible for a substantial percentage of yearly musculoskeletal injuries. With increasing age and inactive lifestyle, the risk for these injuries rises [93].

In response to unphysiological loading, the decreased expression of collagen as well as the upregulation of catabolic enzymes and pro-inflammatory mediators by tenocytes, affect the tendon structure and mechanical stability of ECM and consequently enhances the risk of rupture [94,95]. Tendon injuries are particularly problematic because of dense, specially organized collagenous tissues. The natural healing process is slow and often results in the formation of labile-regenerated tissue and recurrent risk of rupture. Tendons and ligaments are connective tissues with a very high percentage of ECM compared to other tissues [96,97]. Tendons transmit the force from contracting skeletal muscle to bone and have high tensile strength [2].

The resident cell population in tendons are tenocytes, responsible for forming the ECM. The mechanosensitive tenocytes can modify the ECM in response to shifts in the local loading environment. Tendons are predominantly loaded to a great extent along their longitudinal axis in the same direction of skeletal muscle contraction. In these “traction” regions, the collagen fibrils are strictly uniaxial and organized in the same order of muscle force transmission, and the tenocytes align linearly between them. The tenocytes are characterized by the expression of scleraxis (Scx), Mohawk, tenomodulin, etc. The distinct collagen type in the middle of the traction tendon tissue is collagen type I, which is mainly responsible for its high tensile strength [98].

By a wrap around a hypomochlion (bony or fibrous tissue), the tendon changes its direction. In the region of deflection, the additional pressure and shearing stress lead to the formation of fibrocartilaginous tissue in the tendon. The physiological generation of fibrocartilaginous tissue is linked with a high synthesis of collagen type II, which reduces local pressure. In this region, oval-shaped fibrocartilage cells are enclosed between the collagen fibrils. These cells are typically characterized by synthesizing collagen type II and aggrecan, such as hyaline cartilage tissue. The resulting “gliding” tendon is avascular and considered a critical factor for the etiology of degenerative tendon disease [99,100,101,102].

Fibrocartilage areas also exist in the attachment region of tendons or ligaments to the bone, i.e., at their “enthesis”. Enthesis reduces the load in the junction area by distributing the acting biomechanical energy, thus absorbing peak mechanical loads. The local cartilage tissue in the gliding tendon and enthesis area depends on the exerted compressive and tension forces ratio. Expression of the chondrocyte specific-marker SRY-box containing gene 9 (Sox-9) is also detected in enthesis and gliding regions of tendons [103]. It is critical to express the fibrocartilaginous components of the ECM, which are responsible for the high compressive strength.

Both tenocytes and fibrocartilage cells arise from mesodermal compartments. ECM-induced tenogenesis in mesenchymal stem cells (MSCs) is mediated, in particular, by integrins and TGF-β crosstalk, which are affected significantly by the biomechanical environment of cells. Following the binding of TGF-β to its receptor, members of the Smad family are phosphorylated and translocate to the nucleus, where they act as transcription factors to activate the expression of fibrosis-related genes such as a-SMA, Sox-9, Scx, etc. [104,105]. While TGF-β has a moderated role in the activation of Scx, which is vital for the development of traction tendon, it has essential roles in chondrogenesis and the development of fibrocartilage tissues [106,107,108].

The adequate mechanical force induces active TGF-β via integrin αvβ6, which supports Scx expression and is essential to maintain cell function and homeostasis [109]. The increased expression of TGF-ß caused by a higher level of mechanical stress in the form of pressure and/or shear stress leads to the expression of Sox-9 and chondrogenic changes of tenocytes’ phenotype and upregulation of aggrecan, versican, biglycan, and collagen type II in the traction area followed by a decrease of tensile strength [110,111]. The same chondrogenic phenotype is mentored physiologically in gliding and enthesis tendons, with high resistance to pressure and shear stress and less tensile stress [96,112].

The cell–matrix interaction enables tenocytes and fibrocartilage cells to sense and react to mechanical loading with a catabolic or anabolic response. Integrins are transmembrane receptors that anchor cells to the ECM and are integrated into focal adhesions, transferring mechanical force and signals between cells and the surrounding matrix [113,114]. The repetitive stretching of tendons and mechanical stimulation of integrin-β1 in tenocytes activates AKT and mTOR pathways regulating collagen expression, indicating a role of integrins in matrix remodeling. Mechanically sensitive channels react rapidly to mechanical stimuli. In addition, tendon cells express a variety of mechanosensitive ion channels and receptors, including TRPV4 and PIEZO channels and primary cilia [107]. Mechanosensitive TRPV4 and PIEZO channels are the most important ion channels that preferentially conduct calcium (Ca^2+^) in response to biomechanical loading, including stretching, shear stress, and substrate deflection. They are responsible for regulating Ca^2+^ signaling and are essential for cell development, remodeling, and homeostasis of tendon and fibrocartilage tissue [115,116,117,118,119,120].

Tendon overuse causes micro-injuries and tendon inflammation [121,122]. Tendon overuse implies repeated strains of a tendon and excessive mechanical loading. The failing of tendon by overuse is also rendered by fibrocartilage tissue of gliding tendon and enthesis. These areas show metabolic activity, alteration of ECM composition, increasing MMP synthesis, loss of collagen fibers, and lipid accumulation. Maeda et al. used the complete transection model to mimic clinically acute tendon injuries. Their findings demonstrate the critical role of excessive release of active TGF-β during tendon injury and massive tenocyte death. In this study, the temporary lack of tensile loading induces reversible loss of Scx expression, destabilizes the ECM’s structure, and leads to an immoderate release of active TGF-ß [123]. The release of high amounts of TGF-ß caused by excessive loading can lead to angiogenesis and increasing matrix metalloproteases (MMPs) synthesis, the proliferation of fibroblasts, and the development of vascular fibrosis, inflammation, and pain [107]. The tendon inflammation and injuries are associated with oxidative stress, leading to reactive oxygen species and the production of algetics as leukotriens (LTB), histamine, and prostaglandin E2 (PGE2) [124].

According to the study of Zhang et al., TRPV4-mediated Ca^2+^ signaling has a central role in the response of chondrocytes to the physiological stress level. In contrast, PIEZO-mediated Ca^2+^ signaling plays a critical role in the response of cells to overuse mechanical stress [125]. Overstretched ECM triggers the increasing intracellular Ca^2+^ levels and activates the mechanosensitive ion channel PIEZO-1. This activation of PIEZO-1 increases intracellular levels of oxidants and expression of circulating glucose-regulated protein 78 (GRP78) and C/EBP homologous protein (CHOP), which contribute to oxidative stress and endoplasmic reticulum (ER) stress. Moreover, stiff ECM exacerbated oxidative stress-induced senescence and apoptosis in cells [126].

The long-term increase of PGE2 and LTB in response to recurrent mechanical overloading and inflammation leads to tendon degeneration involving deposition of the ECM and collagen fibers, shifting the collagen type III ratio towards collagen type I. This shift increases tendon instability and the risk of rupture.

Some studies show the relived of local reactive oxygen species accumulation in injured tissue using an application of antioxidants, which promote the synthesis of endogenous glutathione to inhibit oxidative stress [78,127,128]. Recently, Lu et al. explored the effects of antioxidant N-acetyl-L-cysteine (NAC) on differentiation and functions of tendon stem/progenitor cells in vitro and the repair function of injured tendons in rats in vivo [129].

Increased levels of VEGF are detected during the healing process. Synthesis of VEGF is boosted during the mechanical loading in both tenocytes and chondrocytes [130,131,132]. Our previous studies demonstrated the link between VEGF and Nrf2 [23,133,134,135]. It is assumed that increasing levels of VEGF and the activation of the Nrf2/ARE pathway have a critical role during the tendon healing process [134]. Using Nrf2 activators, such as sulforaphane, the healing process can be improved [134,135]. The synergistic effect of the Nrf2 activator and adequate mechanical loading to treat tendinopathy should be investigated.

In conclusion, tenocytes’ physiology and functionality rely on mechanical loading and related signaling pathways. However, the link to oxidants and cognate antioxidant defense machinery is still elusive but highly relevant to understand tenocyte biology, as highlighted. According to the available literature, each of the before-mentioned mechanotransduction pathways (Figure 2), namely ion channels, G protein-coupled receptors, growth factor receptors, and integrins, are involved in force transduction in both traction and gliding of tendons. Even if there is only limited direct evidence, this indicates that biomechanical stimulation of tenocytes might also lead to Nrf2 activation. To pursue this hypothesis, further investigation on that topic is needed. Consequently, there is high demand for basic research on the Nrf2 system in tenocytes to understand the relationship between mechanical loading and this master regulator of cellular homeostasis and integrity. In this regard, it might be considered to use Nrf2 inducers as a potential therapy for tendinopathy caused by a mechanical imbalance of the tendon. Interestingly, mechanotherapy (i.e., low-level exercise or extracorporeal shock waves) promotes tendon regeneration and provides a reliable route for appropriate postoperative management. However, the role of the Nrf2 system is not clarified in this regard but should be addressed in the future to further improve and strengthen therapeutic applications to tendinopathies.

## 6. Mechanobiology and the Nrf2-Keap1 System in Cartilage

At embryonic and postnatal stages, cartilage plays an essential role in skeletal development and growth and ensures smooth movement of articulating joints throughout life. Thus, dysfunction of cartilage causes a variety of important skeletal disorders.

Cartilage mechanobiology was recently reviewed [136]. Basically, the anisotropic and highly hierarchically structured tissue organization of articular cartilage is designed to withstand mechanical loads [137]. Many of the cellular mechanosensing mechanisms that exist in the other musculoskeletal tissues also exist in chondrocytes, such as membrane channels, integrins, and the cytoskeleton (Figure 2). Skeletal loading exerts complex gradients of mechanical stresses and subsequently generates biochemical signals, which ultimately have profound impacts on cellular responses in joint tissues, while the molecular mechanisms of these processes are yet incompletely understood [137].

Due to the high cartilage water content, hydrostatic pressures (HP) bear approximately 90% of the applied loads in the cartilage tissue. HP are mechanical forces that impact bone and cartilage alike but at different ranges. Mechanosensation in cartilage starts with the pericellular matrix (PCM), majorly modulating mechanical stress, osmotic pressure, and fluid flow in these cells. Though significantly stiffer than the cell itself, the PCM is softer than the ECM microenvironment surrounding chondrocytes. Both PCM and territorial cellular matrix (TCM) have a natural role as mechanoresponsive growth factor reservoirs, to be released by mechanical deformation [137], hence, assuring the bioavailability of growth factors. Other pathways involved in responses to HP include estrogen receptor α-mediated activation of c-Jun N-terminal kinases (JNK) and increased transforming growth factor receptor (TGFR) I activation [137].

Although little is known about redox-related processes and osmolarity in skeletal tissues, Deng et al. propose that hyperosmolarity-induced oxidative stress markers in other cell types, such as corneal epithelial cells [138]. Moreover, Xu and colleagues observed a similar hyperosmolarity-related effect in nucleus pulposus cells [139]. In addition, osmolarity regulates calcium-dependent Sox9 expression [140] and, consequently, chondrogenic differentiation [141,142] in a TGF-β superfamily-dependent manner [143,144]. Wnt signaling pathways also regulate cartilage development, growth, and homeostasis [145]. Rodent studies revealed that β-catenin-dependent signaling is required for endochondral ossification and growth of axial and appendicular skeletons, while excessive activation may cause severe inhibition of cartilage formation [145]. Different studies identified H_2_O_2_ as a potent modulator of Wnt/β-catenin signaling [146,147].

Several biomechanical stimuli are reported to influence chondrocyte integrity; in particular, ion channels appear to be essential for mechanotransduction in these cells [140]. Cell deformation activates membrane channels, such as the Piezo and TRPV family, causing a calcium ion influx into the cell. Both TRPV4 or Piezo1 potentially activate Nrf2 and reduce mitochondrial ROS, most likely superoxide [148]. Interestingly, mechanical loading possesses the ability to trigger glutathione-mediated stress resistance in cartilage, indicating a link between mechanotransduction and redox signaling [149]. *Nrf2* depletion potentially causes compromised mechanical properties in bony tissues [33]. In addition, Wruck and colleagues provide evidence that oxidative distress causes cartilage degradation in a murine antibody-induced rheumatoid arthritis model and indicates that the presence of a functional Nrf2 gene is a major requirement for limiting cartilage destruction [150]. These studies demonstrate a potential link between mechanotransduction, redox status, and subsequent Nrf2 activity in cartilage. However, a cautious interpretation of data may be advised as an interrelation with other major cell stress conditions, such as endoplasmic reticulum stress-induced cell death [151], cannot always be excluded as potentially confounding factors when linking oxidative stress to mechanobiological events.

While limited direct evidence exists for a distinct link between mechanotransduction and Nrf2/ARE signaling pathways in cartilage, indirect evidence points to an existing cross-talk between both cell signaling routes. This warrants further research in the future.

## 7. Concluding Remarks

In this review, we have highlighted the relevance of mechanotransduction in tissues of the musculoskeletal system, i.e., skeletal and cardiac muscles, bones, cartilage, ligaments, and tendons. In addition, we emphasize the significance of the master antioxidant defense regulator Nrf2-Keap1 system in the control of musculoskeletal tissues. Finally, although currently only minor direct evidence exists, the identified indirect links bridge the gap between these two fundamental tissues- and organs-controlling focal points, mechanotransduction, and the Nrf2-Keap1 system, to point towards novel research that should systematically assemble those to gain a detailed understanding of the interplay between mechanical cues on the one side and cellular redox homeostasis on the other side. In future research, it will be important to describe the precise molecular mechanisms and identify Nrf2 target genes that enable Nrf2 to adapt the musculoskeletal system to its changing loads. If these mechanisms can be elucidated, it may be possible to specifically manipulate the Nrf2 system to treat musculoskeletal degenerations such as sarcopenia or cancer cachexia.

Since many of these indirect links are associated with redox-associated research, it has to be mentioned that this consequently comes up with distinct limitations and issues. The correct understanding of terminology and methodology in this field is fundamental for the interpretation of redox-relevant processes. Unfortunately, investigations in the field of oxidative stress research are often affected by the use of improper terminology and/or invalid methodology [152]. We want to emphasize that we identified this circumstance also in a series of the cited studies, leading to potentially biased interpretation of the data by authors due to misuse of terminology and/or the use of inappropriate detection methods. This especially includes the use of imprecise or invalid markers for the detection of oxidative stress or oxidative damage, such as fluorescent probes (e.g., H_2_DCF-DA), malondialdehyde, or the induction of antioxidant enzymes, to name the most common examples. Nevertheless, the herein cited work is generally of good quality, wherefore we are convinced that the knowledge gained therein contributes to the objective of this review. Furthermore, these cited studies allow for a first understanding of whether a possible link between mechanotransduction and redox-mediated Nrf2 activity does exist. However, and for the reasons mentioned, the current findings in the literature regarding oxidative stress should always be critically reviewed and evaluated. For further reading regarding proper terminology and so-far identified valid biomarkers, we recommend the publication series “Biomarkers of oxidative stress studies (BOSS)” [153,154,155,156,157,158] published in the journal Free Radical Biology and Medicine, which pursues the goal of identifying valid oxidative stress biomarkers.

## Figures and Tables

**Figure 1 ijms-24-07722-f001:**
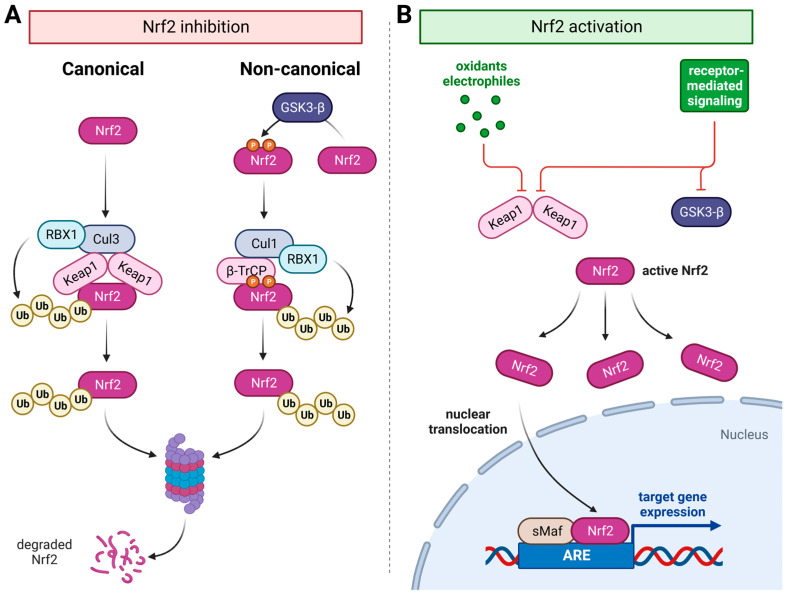
Regulatory network of Nrf2 activity. (**A**) Mechanisms of Nrf2 inhibition via the canonical Keap1-mediated and the non-canonical GSK3-β/β-TrCP-mediated proteasomal degradation. (**B**) Simplified scheme of Nrf2 activation via classical oxidant/electrophile-mediated Keap1 inhibition and/or receptor-mediated Keap1 as well as GSK3-β inhibition. (created with BioRender.com).

**Figure 2 ijms-24-07722-f002:**
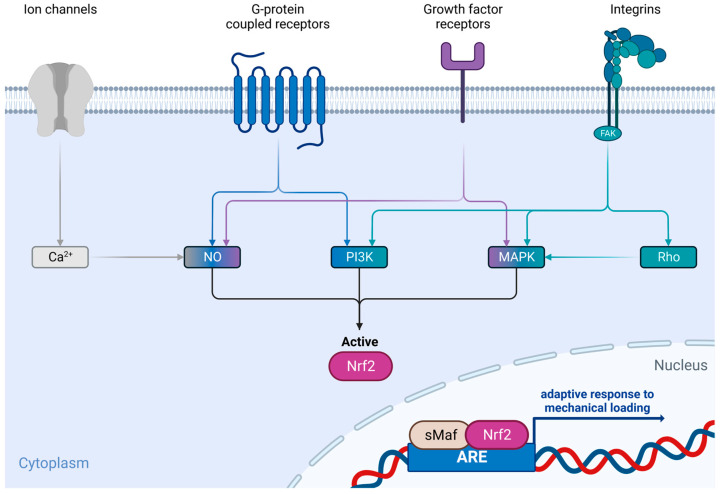
Known mechanisms of mechanosensation and possible downstream cascades that induce the Nrf2/ARE system. (created with BioRender.com).

**Figure 3 ijms-24-07722-f003:**
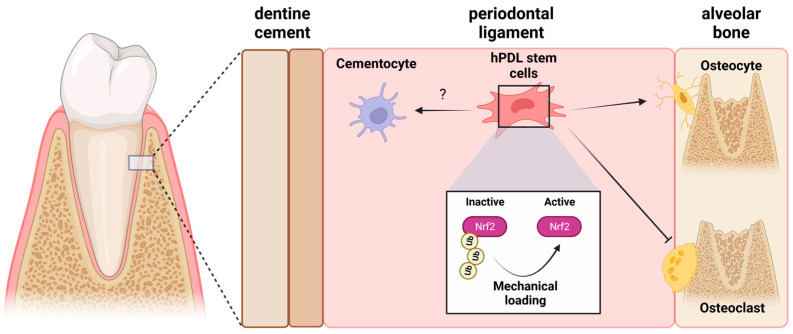
Impact of mechanical loading of periodontal ligament stem cells (hPDLs) on alveolar bone development/maintenance and the potential role of Nrf2 activation in this context. (created with BioRender.com).

## Data Availability

All information may be obtained by contacting the corresponding authors.

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
