# Peer review of "The Contribution of the Nrf2/ARE System to Mechanotransduction in Musculoskeletal and Periodontal Tissues"

_ijms, 2023, doi:10.3390/ijms24097722_

Round 1

Reviewer 1 Report

Fragoulis, A et al have reviewed and discussed the regulation of  Nuclear Factor-E2-related factor 2/Antioxidant response element (Nrf2/ARE) system (a major cellular regulator of exogenous and endogenous oxidative stresses) in structurally delicate tissues to explore its function in their pathophysiology. The authors discussed the pertinence of Nrf2/ARE regulatory functions maintaining signalings in these tissues in the context of reactive oxygen species (ROS) and/or redox biology.

In this study, the authors managed to demonstrate and discuss the highlight that is the relevance of Nrf2/ARE system, a master antioxidant defense regulator in the physiology and pathology of mechanosensitive musculoskeletal tissues and organs like bones, alveolar bones, cartilage, skeletal & cardiac muscles, and tendons. The novelty of this manuscript is the discussion of the regulatory function of the Nrf2/ARE complex in controlling cellular mechanisms ensuring cellular and tissue health.

Although the article provides a balanced view, the authors should discuss several other points with proper references What is oxidative eustress? and how it could influence Nrf2 signaling in muscle tissue.  Or what are canonical & non-canonical pathways and why they are important to Nrf2 activity? (https://pubmed.ncbi.nlm.nih.gov/29913224/) (https://www.ncbi.nlm.nih.gov/pmc/articles/PMC5654572/).

The figures are much appreciated. But 1 of the figure (Figure 3) seems unnecessary to explain sample tissues. One of the major problems is several statements were baseless as they are not supported by appropriate references relevant to this study.

Nonetheless, the article seemed to possess no major concern. Overall, the clarity of the text is good. The manuscript has typographical and grammatical errors which need to be corrected. Figures are much appreciated. In general, the manuscript can accomplish the caliber of quality for consideration for publication in the journal “International Journal of Molecular Sciences” but after further revision. The authors are advised to consider the comments below:

Comments

1.      Page 2 / line 49 / Please provide proper references to support the statements – “The transcription factor Nrf2 has widely been described as one of the main transcription factors involved in the maintenance and the adaptation of the intracellular redox homeostasis.”

2.      Page 2/ Line 56 / “oxidative eustress” – What does eustress mean? Please provide a discussion of 3-4 lines with proper references (https://www.ncbi.nlm.nih.gov/pmc/articles/PMC7930632/)

3.      Page 2 / line 61 / Please provide proper references to support the statements - Nrf2 exerts its primary function under conditions of oxidative eustresses and is thus essential for functional redox signaling.

4.      Figure 1. Regulatory network of Nrf2 activity – A recent review had been discussed about this function (Activators and Inhibitors of NRF2: A Review of Their Potential for Clinical Development, 2019) I am wondering about the novelty.

5.      Please provide a 2-3 line discussion on canonical & non-canonical pathways and why they are important to Nrf2 activity.

6.      I think Figure 3 is unnecessary as authors could explain their intentions or aim in the last paragraph of their introduction precisely and coherently.

7.      Page 4/ line 118 / Please provide proper references to support the statements – For instance, chronically reduced mechanical load causes loss of skeletal muscle tissue and its structure as well as metabolic dysfunction.

8.      A schematic representation of the “Nrf2/ARE system in periodontal alveolar remodeling” would be highly appreciable.

9.      Abstract /Page 1/  line 16 / “an” before essential

10.   Abstract / Page 1/  line 16 / “roles” should change to “role”

11.   Abstract / Page 1/  line 19 / “these tissue” should change to “these tissues”

12.   Introduction / Page 1/ line 37 / The past participle verb known has been used without an auxiliary verb. Consider adding one.

13.   Introduction / Page 2/ line 58 / check spelling “disstress”

14.   Introduction / Page 3/ line 100 / The indefinite article, an, may be redundant when used with the uncountable noun inactivation in your sentence.

15.   Mechanobiology and the Nrf2/ARE system in striated muscles / Page 5/ Line 143 / “ enhanced” It seems that preposition use may be incorrect here, should be “by enhanced”

16.   Mechanobiology and the Nrf2/ARE system in striated muscles / Page 5/ line 153 / check spelling “disstress”

17.   Mechanobiology and the Nrf2-Keap1 system in cartilage / Page 13/ line 556 / check spelling “disstress”

18.   Concluding remarks / Page 13 / line 573 / “these two fundamental tissue” - It appears that the plural demonstrative these is modifying the singular noun tissue. It should be “these two fundamental tissues- and organs”

Author Response

Reviewer 1:

Fragoulis, A et al have reviewed and discussed the regulation of  Nuclear Factor-E2-related factor 2/Antioxidant response element (Nrf2/ARE) system (a major cellular regulator of exogenous and endogenous oxidative stresses) in structurally delicate tissues to explore its function in their pathophysiology. The authors discussed the pertinence of Nrf2/ARE regulatory functions maintaining signalings in these tissues in the context of reactive oxygen species (ROS) and/or redox biology.

In this study, the authors managed to demonstrate and discuss the highlight that is the relevance of Nrf2/ARE system, a master antioxidant defense regulator in the physiology and pathology of mechanosensitive musculoskeletal tissues and organs like bones, alveolar bones, cartilage, skeletal & cardiac muscles, and tendons. The novelty of this manuscript is the discussion of the regulatory function of the Nrf2/ARE complex in controlling cellular mechanisms ensuring cellular and tissue health.

Although the article provides a balanced view, the authors should discuss several other points with proper references What is oxidative eustress? and how it could influence Nrf2 signaling in muscle tissue.  Or what are canonical & non-canonical pathways and why they are important to Nrf2 activity? (https://pubmed.ncbi.nlm.nih.gov/29913224/) (https://www.ncbi.nlm.nih.gov/pmc/articles/PMC5654572/).

The figures are much appreciated. But 1 of the figure (Figure 3) seems unnecessary to explain sample tissues. One of the major problems is several statements were baseless as they are not supported by appropriate references relevant to this study.

Nonetheless, the article seemed to possess no major concern. Overall, the clarity of the text is good. The manuscript has typographical and grammatical errors which need to be corrected. Figures are much appreciated. In general, the manuscript can accomplish the caliber of quality for consideration for publication in the journal “International Journal of Molecular Sciences” but after further revision. The authors are advised to consider the comments below:

Comment 1: 1. Page 2 / line 49 / Please provide proper references to support the statements – “The transcription factor Nrf2 has widely been described as one of the main transcription factors involved in the maintenance and the adaptation of the intracellular redox homeostasis.”

Response: We thank the reviewer for this valuable comment. We added a very useful and  comprehensive review written by Sies and colleagues (2017, Annual Review of Biochemistry), which is a very sophisticated summary of what is known about oxidative stress and related topics. We hope that this will satisfy the reviewer.

Comment 2: Page 2/ Line 56 / “oxidative eustress” – What does eustress mean? Please provide a discussion of 3-4 lines with proper references (https://www.ncbi.nlm.nih.gov/pmc/articles/PMC7930632/)

Response: We appreciate the reviewer’s recommendation to add more information on this important term of “oxidative eustress” since it might be not familiar to all readers. We therefor added the requested 3-4 lines as well more references including the suggested one.

Comment 3: Page 2 / line 61 / Please provide proper references to support the statements - Nrf2 exerts its primary function under conditions of oxidative eustresses and is thus essential for functional redox signaling.

Response: We agree with the reviewer that this statement needs a more precise formulation, we therefore rephrased the sentence and referred once again to the publication of Sies and Ursini in which this statement is nicely summarized in Table 1.

Comment 4: Figure 1. Regulatory network of Nrf2 activity – A recent review had been discussed about this function (Activators and Inhibitors of NRF2: A Review of Their Potential for Clinical Development, 2019) I am wondering about the novelty.

Response: We understand the point brought up by the reviewer that the regulatory network of Nrf2 in general has already been shown and reviewed elsewhere. However, we believe that such an introductionary figure might help the reader to get into the topic.

Comment 5: Please provide a 2-3 line discussion on canonical & non-canonical pathways and why they are important to Nrf2 activity.

Response: We rephrased and added more information on this on page 3 lines 87-90.

Comment 6: I think Figure 3 is unnecessary as authors could explain their intentions or aim in the last paragraph of their introduction precisely and coherently.

Response: This is a good point. We agree with the reviewer and decided to exclude the figure from the manuscript and use it as graphical abstract instead.

Comment 7: Page 4/ line 118 / Please provide proper references to support the statements – For instance, chronically reduced mechanical load causes loss of skeletal muscle tissue and its structure as well as metabolic dysfunction.

Response: Absolutely an important point. To strengthen this statement, we include 3 more references now on page 4 line 127.

Comment 8: A schematic representation of the “Nrf2/ARE system in periodontal alveolar remodeling” would be highly appreciable.

Response: We really thank the reviewer for this valuable suggestion. We now included a new figure (now Figure 3), which summarizes the current knowledge about Nrf2 activity in PDL stem cells and the impact on alveolar bone development and maintenance.

Comment 9-18: Minor issues with spelling or grammar.

Response: We addressed all of the mentioned issues and changed the text accordingly.

Reviewer 2 Report

The manuscript reviewed current knowledge of Nrf2/ARE system to mechanotransduction in musculoskeletal and periodontal tissues including  striated muscles, periodontal alveolar remodeling, tendons and cartilage.  Overall, the manuscript is well organized and well written. 

Too many paragraphs look a little scattered. Please re-organized the text according to the major focus of each section.

Some sentences look weird, please polish the language of the whole manuscript.

Author Response

Reviewer 2:

The manuscript reviewed current knowledge of Nrf2/ARE system to mechanotransduction in musculoskeletal and periodontal tissues including  striated muscles, periodontal alveolar remodeling, tendons and cartilage.  Overall, the manuscript is well organized and well written.

Too many paragraphs look a little scattered. Please re-organized the text according to the major focus of each section.

Some sentences look weird, please polish the language of the whole manuscript.

Response: We thank the reviewer for this very positive evaluation of our manuscript. We highly appreciate this. As suggested, we screened the text once again and additional asked a native speaker for professional support.

Reviewer 3 Report

The stability of transcription factor NRF2 is regulated through Keap1- or GSK3-beta dependent mechanisms. Oxidative eustress and signaling through integrins, ion channels, GPCR and growth factor receptors suppress the Keap-1- and GSK3-beta-dependent degradation of NRF2 in the cytoplasm resulting in its translocation to the nucleus, binding to the ARE response element and expression of proteins responsible for cell protection against oxidative stress. The authors of the review focus on the role of NRF2/ARE system in mechanotransduction in muscle and skeleton. They discuss the available literature on NRF2/ARE signaling in skeletal muscles, cardiomyocytes, periodontal tissues, bone, cartilage and tendons.  The review is quite detailed and the authors present a thoughtful critical discussion of the relevant literature. The illustrations are clear and informative. This is a high-quality review, which will be useful for a wide readership of the journal. The authors, however, should discuss, which specific genes are regulated through NRF2/ARE system in muscles and skeleton and what is the role of these genes in the trophic effects of mechanical stress. In addition, what are the relative inputs of ROS, receptors, integrins and ion channels in the stabilization of NRF2 in different elements of the muskuloskeletal system?

Author Response

Reviewer 3:

The stability of transcription factor NRF2 is regulated through Keap1- or GSK3-beta dependent mechanisms. Oxidative eustress and signaling through integrins, ion channels, GPCR and growth factor receptors suppress the Keap-1- and GSK3-beta-dependent degradation of NRF2 in the cytoplasm resulting in its translocation to the nucleus, binding to the ARE response element and expression of proteins responsible for cell protection against oxidative stress. The authors of the review focus on the role of NRF2/ARE system in mechanotransduction in muscle and skeleton. They discuss the available literature on NRF2/ARE signaling in skeletal muscles, cardiomyocytes, periodontal tissues, bone, cartilage and tendons.  The review is quite detailed and the authors present a thoughtful critical discussion of the relevant literature. The illustrations are clear and informative. This is a high-quality review, which will be useful for a wide readership of the journal.

Comment 1: The authors, however, should discuss, which specific genes are regulated through NRF2/ARE system in muscles and skeleton and what is the role of these genes in the trophic effects of mechanical stress.

Response: We agree with the reviewer that this might be an important point. As also stated in the review the current literature only provides minor direct links in this context, so that one could only speculate on that, which we believe would not be absolutely in line with objective evidence-based science. However, because of the importance of it, we took up this aspect as kind of outlook for future research within our concluding remarks.

Comment 2: In addition, what are the relative inputs of ROS, receptors, integrins and ion channels in the stabilization of NRF2 in different elements of the muskuloskeletal system?

Response: This would definitely be an interesting and valuable information, however, at least to our knowledge there is no experimental data, which would reflect this relevant information in meaningful values. We apologize that this response might not be as satisfying as requested.

Round 2

Reviewer 3 Report

The authors properly answered the critiques.